# IsarStep: a Benchmark for High-level Mathematical Reasoning

**Wenda Li**
University of Cambridge
wl302@cam.ac.uk

**Lei Yu**
DeepMind
leiyu@google.com

**Yuhuai Wu**
University of Toronto, Vector Institute
ywu@cs.toronto.edu

**Lawrence C. Paulson**
University of Cambridge
lp15@cam.ac.uk

## Abstract

A well-defined benchmark is essential for measuring and accelerating research progress of machine learning models. In this paper, we present a benchmark for high-level mathematical reasoning and study the reasoning capabilities of neural sequence-to-sequence models. We build a non-synthetic dataset from the largest repository of proofs written by human experts in a theorem prover. The dataset has a broad coverage of undergraduate and research-level mathematical and computer science theorems. In our defined task, a model is required to fill in a missing intermediate proposition given surrounding proofs. This task provides a starting point for the long-term goal of having machines generate human-readable proofs automatically. Our experiments and analysis reveal that while the task is challenging, neural models can capture non-trivial mathematical reasoning. We further design a hierarchical transformer that outperforms the transformer baseline. The dataset and models are available from: `https://github.com/Wenda302/IsarStep`.

## 1 Introduction

Neural networks have achieved outstanding performance on a wide range of problems in natural language processing, computer vision, and speech recognition. However, research investigating their capacity of doing mathematical reasoning is still limited, with earlier attempts focusing on simple arithmetic tasks like integer addition and multiplication (Zaremba & Sutskever, 2014; Kaiser & Sutskever, 2016; Trask et al., 2018). More recently, there has been work on solving school-level mathematical problems (Saxton et al., 2019), logical reasoning (Evans et al., 2018), and problems of function integration, ordinary differential equations (Lample & Charton, 2020), and properties of differential systems (Charton et al., 2020). While these are valuable contributions to the machine learning community, they focused on generating answers to questions from a specific domain and were carried out on synthetic datasets with small vocabulary (e.g. up to 100 unique tokens).

In this paper, we consider general undergraduate and research-level mathematical proofs as a target for neural networks. When humans prove a theorem, a crucial step is to propose an intermediate proposition to bridge the gap between the goal and the currently known facts. This step requires complicated reasoning capabilities such as creative thinking, inference, understanding existing conditions, and symbolic manipulation of rules. For example, consider the following proof of the irrationality of $\sqrt{2}$:

*Proof of irrationality of $\sqrt{2}$.* Assume $\sqrt{2}$ is rational. Then there exists a pair of coprime integers $a$ and $b$ such that $\sqrt{2} = a/b$, and it follows that $2 = a^2/b^2$ and then $2b^2 = a^2$. Hence $a$ is even. Thus there exists an integer $c$ such that $a = 2c$, which combined with $2b^2 = a^2$ yields $2c^2 = b^2$: hence $b$ is also even. So $a$ and $b$ are both even although they are coprime, contradiction. □

```
1  theorem "sqrt 2 ∉ ℚ"
2  proof
3    assume "sqrt 2 ∈ ℚ"
4    then obtain a b :: int where "sqrt 2 = a/b" "coprime a b"
5      by (metis Rat_cases Rats_def imageE normalize_stable of_rat_divide
6            of_rat_of_int_eq quotient_of_Fract quotient_of_div)
7    then have "2 = a² / b²" by (smt of_int_power power_divide real_sqrt_pow2)
8    then have *:"2*b² = a²"
9      by (cases "b=0",auto simp:field_simps,use of_int_eq_iff in fastforce)
10   then have "even a" by (metis dvd_triv_left even_mult_iff power2_eq_square)
11   then obtain c::int where "a=2*c" by blast
12   with * have "b² = 2*c²" by simp
13   then have "even b" by (metis dvd_triv_left even_mult_iff power2_eq_square)
14   with ‹even a› ‹coprime a b› show False by auto
15 qed
```

Figure 1: Full declarative proof the irrationality of $\sqrt{2}$ in Isabelle/HOL.

To derive $\exists c \in \mathbb{Z}. \, a = 2c$ from $2b^2 = a^2$, the intermediate proposition "$a$ is even" would reduce the gap and lead to a successful proof. We would like to simulate the way humans prove theorems by proposing an intermediate proposition synthesis task — IsarStep. Instead of having primitive steps like $3 + 5 = 8$, the proof steps in IsarStep are at a higher-level, with much bigger steps as basic. Therefore it usually cannot be simply solved by pattern matching and rewriting. To succeed in this task, a model is required to learn the meaning of important mathematical concepts (e.g. determinant in linear algebra, residue in complex analysis), how they are related to each other through theorems, and how they are utilised in proof derivations. Solving the IsaStep task will be potentially helpful for improving the automation of theorem provers, because proposing a valid intermediate proposition will help reduce their search space significantly. It is also a first step towards the long-term goal of sketching complete human-readable proofs automatically.

We have built the IsarStep dataset by mining arguably the largest publicly-hosted repository of mechanised proofs: the Achieve of Formal Proofs (AFP).[1] The AFP is checked by the Isabelle proof assistant (Paulson, 1994) and contains 143K lemmas. Combining the AFP with the standard library of Isabelle/HOL yields a dataset of 204K formally-proved lemmas. The dataset covers a broad spectrum of subjects, including foundational logic (e.g. Gödel's incompleteness theorems), advanced analysis (e.g. the Prime Number Theorem), computer algebra, cryptographic frameworks, and various data structures. A nice property of the mined formal proofs is that they are mostly *declarative proofs*, a proof style very close to human prose proofs.[2] Fig. 1 illustrates the proof of irrationality of $\sqrt{2}$ in Isabelle. We can see that the proof is actually legible (even to people who are not familiar with the system) and and it captures high-level structures like those in human proofs.

We further explore the reasoning capabilities of neural models. We frame the proposed task as a sequence-to-sequence (seq2seq) prediction problem. Beyond evaluating the existing neural seq2seq model baselines—the seq2seq with attention (Bahdanau et al., 2015), the transformer (Vaswani et al., 2017)—we also propose a new architecture, the hierarchical transformer (§4). The architecture is motivated by the way humans reason about propositions; it consists of a set of local transformer layers, modelling the representation of each proposition, and a set of global layers, modelling the correlation across propositions. Experiments (§5) show that these neural models can solve 15–25% of problems on the test set, and the hierarchical transformer achieves the best result. Further analysis (§6) on the output of these models shows that while the proposition synthesis task is hard, the neural models can indeed capture mathematical reasoning. We find that the embeddings of closely related mathematical concepts are close in cosine space; models can reason about the relation between set, subset, and member, and perform more complex multi-step reasoning that is even hard for humans.

Our contributions are summarised as follows:

1. We mine a large non-synthetic dataset of formal proofs and propose a task for evaluating neural models' mathematical reasoning abilities. The dataset contains 820K training examples with a vocabulary size of 30K.

---

[1] https://www.isa-afp.org
[2] A comparison of proofs in different systems is available in Wiedijk (2006). The declarative style proof is also available in Mizar (Grabowski et al., 2010), where the style originates.

2. We evaluate existing neural seq2seq models on this task.

3. We introduce a hierarchical transformer model, which outperforms the baseline models.

4. We provide a comprehensive analysis of what has been learned by the neural models.

5. We provide a test suite to check the correctness of types and the validity of the generated propositions using automatic theorem provers.

## 2 THE ISARSTEP TASK

In this section, we define the task of intermediate proposition generation more concretely. We again take the proof of irrationality of $\sqrt{2}$ as an example. We will have the following derivation:

$$\underbrace{2b^2 = a^2}_{(1)} \Rightarrow \underbrace{a \text{ is even}}_{(2)} \Rightarrow \underbrace{\exists c \in \mathbb{Z}.\, a = 2c}_{(3)}.$$

In our proposed task, we would like to generate $(2)$ given $(1)$ and $(3)$. When humans prove a theorem, they implicitly assume certain background knowledge, as lemmas. For example, in this case we assume that we can trivially prove $(1) \Rightarrow (2)$ based on the fact that the product of two numbers are even iff at least one of them is even. In Isabelle (Paulson, 1994), these relevant lemmas (e.g. `even_mult_iff: even (?a * ?b) = (even ?a ∨ even ?b)` corresponding to line 10 in Fig. 1) can be found automatically by its built-in automation Sledgehammer (Blanchette et al., 2011). In our task, we optionally provide these lemmas as extra information in addition to $(1)$ and $(3)$.

The derivation of $(2) \Rightarrow (3)$ in the proof above is a simple step, because only $(2)$ is needed to arrive at $(3)$. In most cases, multiple propositions have to be used together in order to infer a proposition, for example $P_1, P_2, P_3 \Rightarrow P_4$. For these more general cases, we also include the additional propositions (e.g. $P_2$ and $P_1$) as part of the source propositions.

To summarize, each example in the IsarStep dataset is formed by five parts:

**F.1** a target proposition (e.g. $a$ is even),

**F.2** a set of used local propositions to derive **F.1** (e.g. $2b^2 = a^2$),

**F.3** a local proposition derived from the target proposition **F.1** ($\exists c \in \mathbb{Z}.\, a = 2c$),

**F.4** other local propositions and (library) lemmas used to justify **F.3**,

**F.5** a set of used (library) lemmas to justify **F.1** (e.g. `even_mult_iff: even (?a * ?b) = (even ?a ∨ even ?b)`).

We want to synthesise **F.1** given **F.2** – **F.4** with **F.5** optional: the named lemmas in **F.5** are common knowledge and can be used as additional hints. The propositions are generated as a sequence of tokens and therefore the search space is $\Sigma^*$: search over 30K actions (§3.3, vocabulary size for seq2seq models) at every timestep without a predefined maximum output length.

IsarStep can be considered as single step reasoning, which can be repeated to sketch more complex proofs. Good performance on this task is a crucial step for designing models that can automatically prove theorems with minimal human assistance.

## 3 DATASET PREPROCESSSING AND STATISTICS

The mined raw dataset has long propositions and a large number of unique tokens. To alleviate the performance deterioration of machine learning models due to the aforementioned problems, we propose tricks to preprocess the raw dataset, including free variable normalisation and removing unnecessary parentheses. These tricks substantially reduce the sequence lengths and vocabulary size.

### 3.1 THE LOGIC AND TOKENS

The core logic of Isabelle/HOL is simply-typed $\lambda$-calculus with de Bruijn indices for bound variables (Wenzel, 2020, Chapter 2.2). A local proposition or a (library) lemma/theorem is essentially a *term*

in the calculus. As types can be inferred automatically, we drop types in terms (to reduce the size of the vocabulary) and encode a term as a sequence of tokens that include lambda term constructors: `CONST`, `FREE`, `VAR`, `BOUND`, `ABS` (function abstraction), and `$` (function application). Additionally, parentheses have been used in the sequence to represent the tree structure. To give an example, we encode the proposition `even a` as the following sequence of tokens separated by a white space:

```
CONST HOL.Trueprop $ (CONST Parity.semiring_parity_class.even $ FREE <X0>)
```

where `CONST HOL.Trueprop` is a boilerplate function that converts from type `bool` to `prop`; `CONST Parity.semiring_parity_class.even` is the `even` predicate; `FREE <X0>` encodes the Skolem constant `a` in `even a`. Since `a` is a user-introduced local constant that can be arbitrary, we normalised it to the algorithmically generated name `<X0>` in order to reduce the vocabulary size (see §3.2).

Overall, every local proposition and library lemma/theorem is encoded as a sequence of tokens, and can be mostly decoded to the original term with type inference.

## 3.2 FREE VARIABLE NORMALISATION

Due to Isabelle's use of de Bruijn indices, bound variables have already been normalised: $\forall x. P \ x$ is no different from $\forall y. P \ y$, as both $x$ and $y$ are encoded as `BOUND 0`. However, arbitrary variable names can be introduced by the command **fix** in declarative proofs or unbounded variables in lemma statements (e.g. `False ⟹ P` and `False ⟹ Q` are semantically equivalent but with different free variables). To reduce the vocabulary size here, we normalised these free variables like the bound ones. For example, `False ⟹ P` would be normalised to `False ⟹ <V0>` as `P` is the first free variable in the proposition. Such normalisation reduced the vocabulary size by *one third*. The normalisation preserves the semantics, and we can always parse a normalised term back under a proper context.

## 3.3 STATISTICS

We have mined a total of 1.2M data points for IsarStep. We removed examples in which the length of the concatenation of the source propositions, i.e. **F.2** – **F.4** in §2, longer than 800 and the length of the target propositions, i.e. **F.1** in §2, longer than 200, which results in approximately 860K examples. From these examples we randomly sampled 10K examples for validation and test. In the training data, we removed duplicates, the examples whose target propositions exist in the held-out set, and those that are from the same theorems as the propositions in the held-out set. The final dataset split is 820K, 5000, 5000 for the training, validation, and test sets, respectively. The vocabulary size is 29,759.

## 4 MODEL

We define $\boldsymbol{X} = [\boldsymbol{x}^1, \boldsymbol{x}^2, \ldots, \boldsymbol{x}^I]$ as the sequence of $I$ source propositions, and $\boldsymbol{y} = (y_1, y_2, \ldots, y_N)$ as the target proposition containing $N$ tokens. Let $\boldsymbol{x}^i = (x_1^i, x_2^i, \ldots, x_M^i)$ represent the $i$th proposition in the set, consisting of $M$ tokens. Each source proposition $\boldsymbol{x}^i$ belongs to a category **F.2** – **F.4** defined in §2. We annotate the category corresponding to $\boldsymbol{x}^i$ as $\mathcal{C}_i$ and therefore the sequence of categories corresponding to $\boldsymbol{X}$ is $\boldsymbol{\mathcal{C}} = [\mathcal{C}_1, \mathcal{C}_2, \ldots, \mathcal{C}_I]$. The generation of a target proposition $\boldsymbol{y}$ is determined by finding the proposition $\hat{\boldsymbol{y}}$, where $p(\hat{\boldsymbol{y}} \mid \boldsymbol{X}, \boldsymbol{\mathcal{C}})$ is optimal,

$$\hat{\boldsymbol{y}} = \arg\max_{\boldsymbol{y}} p(\boldsymbol{y} \mid \boldsymbol{X}, \boldsymbol{\mathcal{C}}). \qquad (1)$$

We propose two approaches to parameterising the conditional probability $p(\boldsymbol{y} \mid \boldsymbol{X}, \boldsymbol{\mathcal{C}})$, which differ in the way of modelling the sequence of source propositions. The first method is simply appending a label to each source proposition indicating their category and then concatenating the source propositions using a special token `<SEP>`, treating the resulting long sequence as the input to a seq2seq model.

Our second approach models the encoding of source propositions hierarchically. As shown in Fig. 2, the encoder has two types of layers. The local layers build the proposition representations by modelling the correlations of tokens within each proposition; the global layers take the proposition

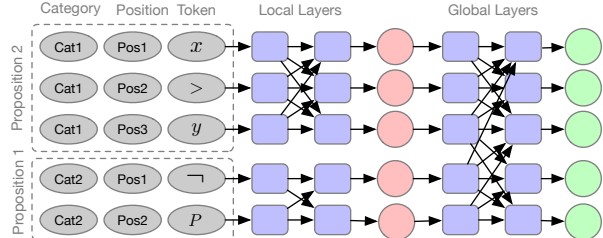

Figure 2: Architecture of the encoder of the hierarchical transformer (HAT). There are two types of layers, the local layers model the correlation between tokens within a proposition, and the global layers model the correlation between propositions. The input to the network is the sum of the token embedding, the positional information, and the embedding of the corresponding category.

Table 1: Test set accurarcies (exact match) and BLEU scores of different models on the IsarStep task.

| Model | Top-1 Acc. | | Top-10 Acc. | | BLEU | |
|---|---|---|---|---|---|---|
| | Base | **+F.5** | Base | **+F.5** | Base | **+F.5** |
| RNNSearch | 13.0 | 16.7 | 26.2 | 32.2 | 42.3 | 52.2 |
| Transformer | 20.4 | 22.1 | 33.1 | 34.6 | 59.6 | 62.9 |
| HAT | **22.8** | **24.3** | **35.2** | **37.2** | **61.8** | **65.7** |

representations as input and model the correlations across propositions. Both the local layers and global layers are transformer layers (Vaswani et al., 2017). Positional information is encoded separately for different source propositions. That is, suppose $x^1$ has $M$ tokens, then the position of the first token in $x^2$ is not $M+1$ but 1. The embedding of a token $x_m^i$ is obtained by adding the token embedding, the positional information, and the embedding of the category that the proposition $x^i$ belongs to. The category embedding is learnt together with the rest of the network. We call this model the *hierarchical transformer* (HAT). Intuitively, HAT models the structure of the source propositions more explicitly compared to the first approach and therefore should be better at capturing reasoning between source and target propositions. We will validate our hypothesis in §5.

## 5 EXPERIMENTS

We benchmark three models on IsarStep (§2), namely the seq2seq model with attention (RNNSearch) (Bahdanau et al., 2015; Wu et al., 2016), transformer (Vaswani et al., 2017), and hierarchical transformer (HAT). The input to the RNNSearch and the transformer is a concatenation of source propositions (the first parameterisation approach described in §4). We train these models with the same training data and report their performance on test sets. See appendix for experimental setup.

### 5.1 EVALUATION

Widely used metrics for text generation are BLEU score (Papineni et al., 2002) and ROUGE score (Lin, 2004) that measure n-gram overlap between hypotheses and references. Both metrics are not ideal in mathematical proofs since a proposition can be invalid due to one or two incorrect tokens. Therefore, in addition to BLEU score, we also consider exact match between hypotheses and references as our evaluation metric. We report top-1 accuracy and top-10 accuracy. The top-1 accuracy is the percentage of the best output sequences that are correct in the given dataset. The top-10 accuracy is the percentage of target sequences appearing in the top 10 generated sequences.

It is possible that models generate alternative valid propositions that are not exactly the same as the references. We further implemented a test suite to bring the generated propositions back to the Isabelle environment and check their correctness using automatic theorem provers (ATPs).

## 5.2 RESULTS

**BLEU and Exact Match**   Table 1 presents the results of different models for the IsarStep task. Overall, the neural seq2seq models achieve around 13–25% top-1 accuracies and 26–38% top-10 accuracies, which indicates that this task is non-trivial and yet not too difficult for neural networks. Of the three models, the transformer (Vaswani et al., 2017) outperforms the RNNSearch (Bahdanau et al., 2015; Wu et al., 2016) significantly and our HAT performs best. As mentioned in §2, adding **F.5** is optional and is conjectured for better performance due to exploiting used lemmas explicitly. We experimented with both cases and found that adding this extra information indeed leads to further improvement. This is consistent with the scenario when humans prove theorems: if humans are told that certain lemmas are relevant to the current proof, they will use these lemmas and have a better chance of success.

**Alternative Valid Propositions**   We consider an output proposition $P$ as an alternative valid intermediate proposition if 1) $P$ is a well-formed proposition and does not match the ground truth at the surface form; 2) $P$ does not match any substring of the source (to avoid it being a simple copy of **F.3** or an assumption in **F.2**); 3) both **F.2** $\Rightarrow$ $P$ and $P$ $\Rightarrow$ **F.3** can be automatically proved by ATPs.[3] Note that this will only give us a lower bound to the number of alternative propositions, due to the limitation of ATPs' automation. Table 2 presents the percentage of correct propositions on the test set. Correct proposition is a proposition

Table 2: Percentage of correct propositions.

| Model | Base | +**F.5** |
|---|---|---|
| Transformer | 25.2 | 26.8 |
| HAT | 27.6 | 29.4 |

that matches either the corresponding ground truth or one of the alternative valid propositions. We can see that alternative propositions contribute 5 percentage point more correct propositions, compared to top-1 accuracy in Table 1.

**Automation Improvement**   In lots of cases, ATPs cannot infer from one step to another automatically (i.e. **F.2** $\Rightarrow$ **F.3**) without the crucial intermediate steps proposed by humans. We found that there are about 3000 cases in our test set that **F.2** $\Rightarrow$ **F.3** cannot be proved automatically by ATPs. And within these 3000 cases, 61 cases can be proved automatically given the generated intermediate propositions from our HAT model: **F.2** $\Rightarrow$ $P$ and $P$ $\Rightarrow$ **F.3**. This is not a big improvement. Further progress is needed to improve seq2seq models' reasoning capability in order to improve the automation of theorem provers significantly.

**Better Generalisation of HAT**   Since the transformer and HAT have different source sequence encoding, we explore how well these two models perform on examples with various source sequence lengths. We categorise the examples on the IsarStep test set into 5 buckets based on their source lengths and calculate the top-1 accuracies for different buckets, as shown in Fig. 3. Interestingly, although we did not train the models with source sequences longer than 512, they can still achieve reasonable accuracies on long sequences. In particular, HAT performs significantly better than the transformer on sequences longer than 480. Especially in the length bucket of 640–800, HAT doubles the accuracy of the transformer.

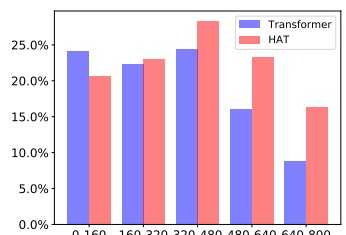

Figure 3: Accuracy of different source sequence lengths.

**Importance of Category Information**   We subsequently investigate the effect of incorporating the category information for source propositions into the models by removing the category embedding for the input to the HAT encoder (Fig. 2), i.e. we are now modelling $p(\boldsymbol{y} \mid \boldsymbol{X})$ instead of $p(\boldsymbol{y} \mid \boldsymbol{X}, \boldsymbol{\mathcal{C}})$. We see a dramatic drop in accuracy: 14.6 versus 22.8 obtained by the HAT with category embedding included, indicating the importance of category information. This is in line with human proofs: without knowing the logical relations between propositions, we do

---

[3]If **F.2** $\Rightarrow$ **F.3** is directly provable via ATPs, a trivial proposition (e.g. $1 = 1$) can be considered as an alternative. This is hard to detect automatically but could still serve as a fair comparison across seq2seq models as long as they can propose a well-formed trivial proposition in such scenario.

not know what proposition is missing. This indicates that the models are not simply doing pattern matching but should have captured some reasoning information.

# 6  QUALITATIVE ANALYSIS

In this section, we present an analysis of what has been learnt by the neural network models. To summarise our findings: 1) the seq2seq models can learn the syntax of propositions correctly; 2) the learned token embeddings are comprehensive in that related mathematical concepts are close in cosine space; 3) manual inspection of the generated propositions reveal that models can learn non-trivial mathematical reasoning and even more complicated multi-step reasoning.

**Token Embeddings**  To investigate whether the seq2seq models have learnt mathematical reasoning, we checked whether the learnt token embeddings were meaningful. We first projected the learnt embeddings for all the tokens in the vocabulary into a three-dimensional space via principal component analysis and chose random tokens and checked their 50 nearest neighbours in cosine distance. We found that the embeddings of related concepts in mathematics were close, indicating that the models have managed to learn the relations between mathematical concepts — the basic step towards reasoning mathematically. For example, in Fig. 4, the neighbours of 'Borel measurable' are mostly measure theory related including 'almost everywhere', 'integrable', and 'null set', while 'arrow' is close to 'isomorphism' (EpiMonoIso.category.iso), 'identity'(Category.partial_magma.ide), and 'inverse arrow'(EpiMonoIso.category.inv), which are concepts in category theory. Additionally, vector arithmetic also seems to connect related mathematical definitions: for example, the three closest tokens next to 'bounded' + 'closed' are 'bounded','closed', and 'compact', where compactness can be alternatively defined as boundedness and closedness (on a Euclidean space).

**Attention Visulisations**  We next investigate how reasoning has been learnt by visualising attentions from transformer (Vaswani et al., 2017). We find that important and related tokens are likely to attend to each other. For example, Fig. 5 illustrates the visulisation of the last layer of the transformer encoder for the source propositions **F.2**: **F.3**: $x_{70} \in x_{39}$ **F.4**: $x_{57} \subseteq x_{39}$. The target proposition generated from the model is $x_{70} \in x_{57}$. The interpretation of those source propositions is that combining with (**F.4**) $x_{57} \subseteq x_{39}$ we would like to infer the intermediate step so that the goal $x_{70} \in x_{39}$ can be proved. The transformer model gives the correct answer $x_{70} \in x_{57}$ which implicitly applied the lemma

$$x \in A, A \subseteq B \vdash x \in B \tag{2}$$

that relates $\in$ and $\subseteq$. On the last self-attention layer of the transformer encoder (Fig. 5), $\in$ and $\subseteq$ attend to each other. Interestingly, the above reasoning seems robust. If we swap $x_{57}$ and $x_{39}$ in **F.4** (i.e., the source is now **F.2**: **F.3**: $x_{70} \in x_{39}$ **F.4**: $x_{39} \subseteq x_{57}$), the answer becomes $x_{70} \in x_{39}$. This totally makes sense since (2) no longer applies (despite that $\in$ and $\subseteq$ still attend to each other similarly as in Fig. 5) and $x_{70} \in x_{39}$ can only be discharged by proving itself.

**Multi-Step Reasoning**  By further inspecting the generated propositions, we find that the model can implicitly invoke multiple theorems as humans normally do. While this property can be found in quite a few examples, here we show one of them due to the limited space. We refer the readers to the appendix for more examples. Given the source **F.2**: $\dim(\mathrm{span}(x_0)) \leq \mathrm{card}(x_2)$ **F.3**: $\mathrm{card}(x_2) = \dim(x_0)$ **F.4**: $\mathrm{card}(x_2) \leq \dim(x_0), \mathrm{finite}(x_2)$, where $\dim$, $\mathrm{span}$ and $\mathrm{card}$ refer to the dimensionality, the span, and the cardinality of a set of vectors, respectively, and the model gives the correct answer $\dim(x_0) \leq \mathrm{card}(x_2)$. Here, $\dim(x_0) \leq \mathrm{card}(x_2)$ is derived by $\dim(\mathrm{span}(x_0)) \leq \mathrm{card}(x_2)$ only if the model has implicitly learned the following theorem $\vdash \dim(\mathrm{span}(S)) = \dim(S)$, while $\dim(x_0) \leq \mathrm{card}(x_2)$ yields $\mathrm{card}(x_2) = \dim(x_0)$ (in conjunction of $\mathrm{card}(x_2) \leq \dim(x_0)$) only if the model has implicitly invoked the antisymmetry lemma $x \leq y, y \leq x \vdash x = y$.

**Failures**  We observe that incorrect propositions are well-formed and plausible propositions but they are usually a copy of parts of the source propositions.

# 7  RELATED WORK

There have been a series of work that evaluates mathematical reasoning abilities of seq2seq models. The tasks that these works attempt to solve include school-level mathematical problems (Ling et al.,

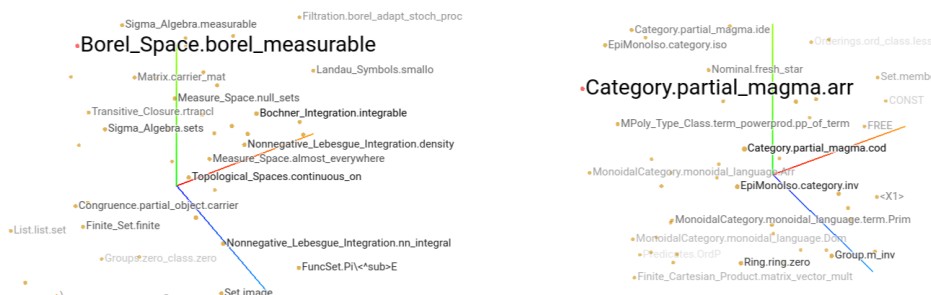

Figure 4: Nearest neighbours of the tokens 'Borel measurable' (left) and 'arrow' (right) in cosine space. The 512-dimensional embeddings are projected into 3-dimensional embeddings. Neighbours are found by picking the top 50 tokens whose embeddings are closest to the selected token.

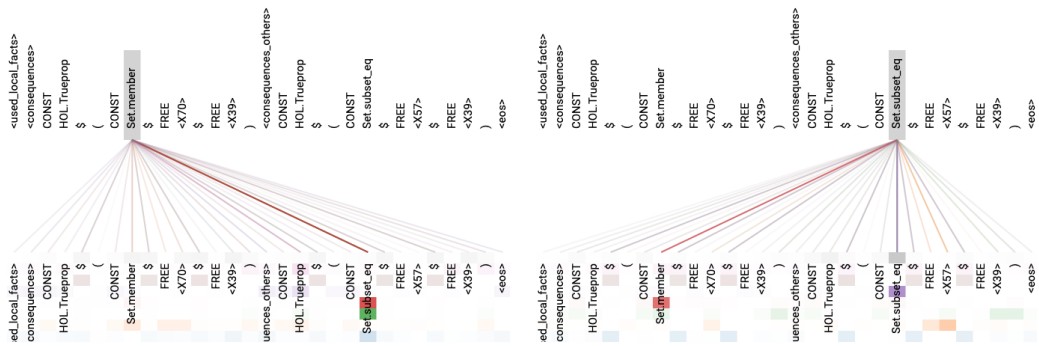

Figure 5: Attention visualisation of the last layer of the transformer encoder for the source propositions **F.2**: **F.3**: $x_{70} \in x_{39}$ **F.4**: $x_{57} \subseteq x_{39}$. The generated target proposition is $x_{70} \in x_{57}$.

2017; Saxton et al., 2019), function integration and ordinary differential equations (Lample & Charton, 2020), properties of differential systems (Charton et al., 2020), SAT formulas and temporal logic (Finkbeiner et al., 2020). Our task is different from the previous ones in the sense that ours is non-synthetic, with realistic vocabulary size (i.e., 30K vs. less than 100) and has a broad coverage topics in research-level mathematics and computer science that have no general algorithmic solutions.

Our work is closely related to the most recent work on applying language modelling to theorem proving. Urban & Jakubův (2020) present initial experiments on generating conjectures using GPT-2 (Radford et al., 2019). Polu & Sutskever (2020) show that the GPT-3 language model (Brown et al., 2020b) additionally pretrained with mathematical equations mined from the web can generate propositions that enable theorem provers to prove more theorems automatically. Rabe et al. (2020) pretrain a masked language models on proofs mined from the HOLIst dataset (Bansal et al., 2019) and apply the pretrained models to the downstream tasks of type inference and predicting conjectures. While both their work and ours find that transformer models have strong mathematical reasoning capabilities, they have different objectives from ours. Their objectives are to show the effectiveness of pretraining on downstream tasks; by contrast we are building benchmarks to test models' ability of solving mathematical problems. In fact, we can pretrain seq2seq models following their proposed methods and verify their effectiveness on our dataset. We will leave this for future work.

There exists a few benchmarks for theorem proving. Kaliszyk et al. (2017) propose a machine learning benchmark for higher order logic reasoning. Alemi et al. (2016) use convolutional networks for *premise selection*. Both of these tasks are classification problems, whereas our proposition generation task is a generation problem with a countably infinite search space. In the benchmarks for *tactic synthesis* (Huang et al., 2019; Bansal et al., 2019; Yang & Deng, 2019; Sanchez-Stern et al., 2019; Paliwal et al., 2019; Gauthier et al., 2017), an agent is asked to propose a sequence of tactics to solve the current goal. Our task is complementary: the model is required to conjecture a goal (intermediate proposition) that is likely to be useful in a derivation. Wu et al. (2020) proposed a synthetic inequality theorem proving benchmark that studies the out-of-distribution generalization abilities of models.

Conjecturing literals in tableau proofs using recurrent neural networks has been investigated by Piotrowski & Urban (2020).

Other related work includes analogy/syntax driven conjecturing (Gauthier et al., 2016; Nagashima & Parsert, 2018; Wang & Deng, 2020), goal classification/ranking (Goertzel & Urban; Brown et al., 2020a), proof method recommendations (Nagashima, 2020; Nagashima & He, 2018), and autoformalisation of mathematics (Wang et al., 2020; Szegedy, 2020).

**Hierarchical Models** Hierarchical models have been proposed to solve natural language processing tasks such as document representation (Yang et al., 2016) and document summarisation (Zhang et al., 2019; Liu & Lapata, 2019). Both our hierarchical transformer (HAT) and those models share the similar spirit of introducing local layers to encode local sentences (or propositions) and global layers to capture cross sentence (or proposition) information. However, our HAT is different from their hierarchical models in the way of representing sentences (or propositions): while their models encode sentences into fixed size vectors, the representation of a proposition in our model is a matrix of dynamic size. The model by Liu & Lapata (2019) has a more sophisticated architecture for capturing sentence representations compared to those by Yang et al. (2016) and Zhang et al. (2019), where they introduce multi-head pooling to encode sentences with different attention weights. Compare to Liu & Lapata (2019)'s model, our model does not introduce additional parameters beyond the standard transformers. Another subtle difference between our model and the existing models is the way of doing positional encoding. Unlike documents where the order of sentences matters, propositions within each category of our IsarStep task do not require an order. Therefore, we do not encode the positional information of different propositions.

## 8 CONCLUSION

We mined a large corpus of formal proofs and defined a proposition generation task as a benchmark for testing machine learning models' mathematical reasoning capabilities. In our defined task, the gap between adjacent proof steps is big and therefore it cannot be simply solved by pattern matching and rewriting. We evaluated the RNN attention model and the transformer on this dataset and introduced a hierarchical transformer that outperforms the existing seq2seq model baselines especially on long source sequences. Our analysis shows that the neural seq2seq models can learn non-trivial logical relations and mathematical concepts. We hope that our work will drive the development of models that can learn to reason effectively and eventually build systems that can generate human-readable proofs automatically.

### ACKNOWLEDGMENTS

Both Li and Paulson are supported by the ERC Advanced Grant ALEXANDRIA (Project 742178), funded by the European Research Council. We would like to thank Dani Yogatama, Adhiguna Kuncoro, Phil Blunsom, and Christian Szegedy for helpful comments on an earlier draft of this paper. We also thank the anonymous reviewers for their insightful suggestions.

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

Table 3: Additional results.

| Model | Top-1 Acc. | | Top-10 Acc. | | BLEU | |
|---|---|---|---|---|---|---|
| | Base | +**F.5** | Base | +**F.5** | Base | +**F.5** |
| Conv2Conv | 8.7 | 8.3 | 21.7 | 20.8 | 48.66 | 46.54 |

Kaiyu Yang and Jia Deng. Learning to Prove Theorems via Interacting with Proof Assistants. In *Proceedings of ICML*, 2019.

Zichao Yang, Diyi Yang, Chris Dyer, Xiaodong He, Alexander J. Smola, and Eduard H. Hovy. Hierarchical attention networks for document classification. In Kevin Knight, Ani Nenkova, and Owen Rambow (eds.), *Proceedings of NAACL HLT*, 2016.

Wojciech Zaremba and Ilya Sutskever. Learning to execute. *arXiv preprint arXiv:1410.4615*, 2014.

Xingxing Zhang, Furu Wei, and Ming Zhou. HIBERT: document level pre-training of hierarchical bidirectional transformers for document summarization. In *Proceedings of ACL*, 2019.

## A EXPERIMENTAL SETUP

For RNNSearch[4] (Bahdanau et al., 2015; Wu et al., 2016), we use 2-layer LSTMs (Hochreiter & Schmidhuber, 1997) with 512 hidden units and 0.2 dropout rate. The hyperparameters for training the transformer[5] are the same as *transformer base* (Vaswani et al., 2017), i.e. 512 hidden size, 2048 filter size, 8 attention heads, and 6 layers for both the encoder and decoder. The hyperparameters for HAT are the same, except that the number of local context layers is 4 and global context layers is 2. We share the source and target token embeddings for all the three models. We use beam search decoding with beam size 5 (for top1 accuracies) and 10 (for top10 accuracies). The configurations for different models are the best ones we found based on validation performance. We train these models for 100K steps and pick the checkpoint with the best BLEU on the validation set to evaluate on the test set. Training the transformer and HAT takes 72 hours on 4 Tesla-V100 GPUs.

## B ADDITIONAL EXPERIMENTAL RESULTS

We report additional experimental results from convolutional seq2seq models (Gehring et al., 2017)[6] in Table 3. We use the setup of `fconv_iwslt_de_en` to train the model.

## C TEST SUITE

We use the default Sledgehammer (Blanchette et al., 2011) method in Isabelle as our automatic theorem prover for checking derivations. To ensure fair and efficient comparisons, we shut off three of its options: "isar_proofs", "smt_proofs" and "learn". The timeout for Sledgehammer is 30s. We run the test suite on a platform with Intel 9700K CPU and 32G RAM, and it takes about 44 hours to evaluate on the test set. Due to some technical issues (e.g., the sampled example appears before Sledgehammer is introduced when booting Isabelle/HOL), 92/5000 examples from the test set are not supported by the current version of our test suite. We present the percentage of well-formed propositions (i.e., outputs that type checks in Isabelle/HOL) of Transformer and HAT in Table 4.

Table 4: Percentage of well-formed propositions

| Model | Base | +**F.5** |
|---|---|---|
| Transformer | 58.2 | 60.1 |
| HAT | 58.2 | 58.9 |

---

[4]Codebase: `https://github.com/tensorflow/nmt`

[5]Codebase: `https://github.com/THUNLP-MT/THUMT/tree/pytorch`

[6]Codebase: `https://github.com/pytorch/fairseq/tree/master/examples/conv_seq2seq`

With such settings, Sledgehammer can automatically prove the goal **F.3** in 1984/4908 examples. By incorporating the ground truth **F.1**, the derivations (i.e., to prove both **F.2** $\Rightarrow$ **F.1** and **F.1** $\Rightarrow$ **F.3**) can be closed automatically in 2243 examples. Among the 1125 examples that HAT produces an exact match, 466 of them have a 'small' gap: Sledgehammer discharges the goal **F.3** directly; 61 of gaps are 'just right': the introduced intermediate step **F.1** can help Sledgehammer bridge the gap; most of the remaining 598 examples have 'large' gaps (in either **F.2** $\Rightarrow$ **F.1** or **F.1** $\Rightarrow$ **F.3**) that are beyond the capability of Sledgehammer. It appears that the insignificant amount of automation improvement can be attributed to the limited number of 'just right' gaps that are within the reach of Sledgehammer.

## D    ALTERNATIVE STEPS

Many alternative steps are trivially equivalent to the ground truth (e.g. $A = B$ given the ground truth being $B = A$, and $P \wedge 1 = 1$ given the truth being $P$). However, we still manage to find a few non-trivial ones, and one of them (#954 in the test set) even identifies a redundant derivation in the Isabelle standard library:

**F.1** :
$$0 = x_1(x_9)2(2\pi)\mathrm{i}n(x_3, x_9) \tag{3}$$

**F.2** :
$$x_7 = \{w \mid w \notin \mathrm{path\_image}(x_3) \wedge n(x_3, w) = 0\} \tag{4}$$
$$x_9 \in x_7 \tag{5}$$

**F.3** :
$$\oint_{x_3} \frac{dx}{x - x_9} = 0 \tag{6}$$

**F.4** :
$$\forall z \notin \mathrm{path\_image}(x_3). \oint_{x_3} \frac{dx}{x - z} = \frac{n(x_3, z)}{2\pi \mathrm{i}} \tag{7}$$
$$x_7 = \{w \mid w \notin \mathrm{path\_image}(x_3) \wedge n(x_3, w) = 0\} \tag{8}$$
$$x_9 \in x_7 \tag{9}$$

Here, $\mathrm{i}$ is the imaginary unit, $\mathrm{path\_image}(x_3)$ returns the image of the contour $x_3$ on the interval $[0, 1]$, and $n(x_3, x_9)$ is the winding number of $x_3$ around the point $x_9$. **F.2** $\Rightarrow$ **F.1**: combining (4) and (5) leads to $n(x_3, x_9)$ that proves (3). **F.1**, **F.4** $\Rightarrow$ **F.3**: joining (8) and (9) yields

$$x_9 \notin \mathrm{path\_image}(x_3) \tag{10}$$

$$n(x_3, x_9) = 0 \tag{11}$$

By further joining (10) with (7) we have

$$\oint_{x_3} \frac{dx}{x - x_9} = \frac{n(x_3, x_9)}{2\pi \mathrm{i}}$$

which leads to (6) considering $n(x_3, x_9) = 0$ (i.e., (11)). Note that our **F.1** is not used in the derivation above hence redundant. Instead of the redundant ground truth, HAT proposed (10) which is clearly a much better intermediate step.

## E    EXAMPLES OF CORRECT SYNTHESISES

In this section, we present some correctly synthesised propositions which will be labelled as **F.1** in each case.

#605 in the validation set:

**F.1** :

$$\text{additive}(\mathcal{A}_{x_1}, \mu_{x_0}) \tag{12}$$

**F.2** :

$$\text{subalgebra}(x_0, x_1) \tag{13}$$

**F.3** :

$$\text{measure\_space}(\mathcal{X}_{x_0}, \mathcal{A}_{x_1}, \mu_{x_0}) \tag{14}$$

**F.4** :

$$\sigma(\mathcal{X}_{x_0}, \mathcal{A}_{x_1}) \tag{15}$$

$$\text{positive}(\mathcal{A}_{x_1}, \mu_{x_0}) \tag{16}$$

$$\text{measure\_space}(v_1, v_0, v_2) = (\sigma(v_1, v_0) \wedge \text{positive}(v_0, v_2) \wedge \text{additive}(v_0, v_2)) \tag{17}$$

Here, $x_0$ and $x_1$ are measure spaces. For a measure space $y$, $\mathcal{X}_y$, $\mathcal{A}_y$, and $\mu_y$ are the three components of $y$ (i.e., $y = (\mathcal{X}_y, \mathcal{A}_y, \mu_y)$), where $\mathcal{X}_y$ is the carrier set, $\mathcal{A}_y$ is a collection of subsets on $\mathcal{X}_y$, and $\mu_y$ is a measure defined on $\mathcal{A}_y$. **F.2** $\Rightarrow$ **F.1**: $x_1$ being a subalgebra of $x_0$ (i.e., (13)) implies $\mathcal{A}_{x_1} \subseteq \mathcal{A}_{x_0}$, so that $\mu_{x_1}$ in $\text{additive}(\mathcal{A}_{x_1}, \mu_{x_1})$ (i.e., $\mu_{x_1}$ is countably addictive on $\mathcal{A}_{x_1}$ which is implied by $x_1$ being a measure space) can be substituted with $\mu_{x_0}$ which yields (12). **F.1**, **F.4** $\Rightarrow$ **F.3**: deriving (14) requires unfolding the definition of measure spaces (i.e., (17)), which requires $v_0$ is a sigma algebra on $v_1$, the measure $v_2$ is non-negative on $v_0$, and $v_2$ is countably additive on $v_0$. Two of the three requirements have already been satisfied by (15) and (16) respectively, while (12) entails the last one and eventually leads to (14).

#2903 in the validation set:

**F.1** :

$$x_{29} \notin \text{path\_image}(x_7) \tag{18}$$

**F.2** :

$$x_{29} \in \text{proots}(x_0) - \text{proots\_within}(x_0, \text{box}(x_1, x_2)) \tag{19}$$

$$\text{path\_image}(x_7) \cap \text{proots}(x_0) = \{\} \tag{20}$$

**F.3** :

$$x_{29} \notin \text{cbox}(x_1, x_2) \tag{21}$$

**F.4** :

$$\text{cbox}(x_1, x_2) = \text{box}(\text{x}_1, \text{x}_2) \cup \text{path\_image}(x_7) \tag{22}$$

$$x_{29} \in \text{proots}(x_0) - \text{proots\_within}(x_0, \text{box}(x_1, x_2)) \tag{23}$$

Here, $\text{path\_image}(x_7)$ is the image of the path function $x_7$ on the interval $[0, 1]$; $\text{proots}(x_0)$ and $\text{proots\_within}(x_0, S)$ are, respectively, the roots of a polynomial $x_0$ and the roots (of $x_0$) within a set $S$; $\text{box}(x_1, x_2) = \{x \mid x_1 < x < x_2\}$ and $\text{cbox}(x_1, x_2) = \{x \mid x_1 \leq x \leq x_2\}$ are (bounded) boxes on an Euclidean space. **F.2** $\Rightarrow$ **F.1**: $x_{29}$ is a root of $x_0$ (by (19)) that does not intersect with the path of $x_7$ (i.e., (20)). **F.1**, **F.4** $\Rightarrow$ **F.3**: combining with (22), (21) is equivalent to $x_{29} \notin \text{box}(\text{x}_1, \text{x}_2) \wedge x_{29} \notin \text{path\_image}(x_7)$, which follows from joining (23) with (18).

#1514 in the validation set:

**F.1** :

$$\frac{x_4(2x_{10})}{x_4(x_{10})} \leq x_9 \tag{24}$$

**F.2** :

$$x_9 = \text{Max}\left\{ \frac{x_4(2y)}{x_4(y)} \mid y \leq x_8 \right\} \tag{25}$$

$$x_{10} \leq x_8 \tag{26}$$

**F.3** :

$$x_4(2x_{10}) \leq x_9 x_4(x_{10}) \tag{27}$$

**F.4** :

$$0 < x_4(x_{10}) \tag{28}$$

**F.2** ⇒ **F.1**: (26) implies

$$\frac{x_4(2x_{10})}{x_4(x_{10})} \in \left\{ \frac{x_4(2y)}{x_4(y)} \mid y \le x_8 \right\},$$

hence (24) by the definition of Max. **F.1**, **F.4** ⇒ **F.3** by arithmetic and the positivity of the denominator (i.e., (28)).

#1222 in the validation set:

**F.1** :

$$|ix_0 + \sqrt{1 - x_0^2}| = 1 \tag{29}$$

**F.2** :

$$|ix_0 + \sqrt{1 - x_0^2}|^2 = 1 \tag{30}$$

**F.3** :

$$\Im(\arcsin(x_0)) = 0 \tag{31}$$

**F.4** :

$$\Im(\arcsin(v_0)) = -\ln(|iv_0 + \sqrt{1 - v_0^2}|) \tag{32}$$

$$e^{-v_0} = 1/(e^{v_0}) \tag{33}$$

**F.2** ⇒ **F.1** by arithmetic. **F.1**, **F.4** ⇒ **F.3**:

$$\Im(\arcsin(v_0)) = -\ln(|iv_0 + \sqrt{1 - v_0^2}|) = -\ln 1 = 0.$$

#35 in the validation set:

**F.1** :

$$x_4 = x_{10}[x_{12}] \tag{34}$$

**F.2** :

$$\forall x. \, x_3 = x_6[x] \wedge x < \mathrm{len}(x_6) \longrightarrow x_4 = x_{10}[x] \tag{35}$$

$$x_3 = x_6[x_{12}] \tag{36}$$

$$x_{12} < \mathrm{len}(x_6) \tag{37}$$

**F.3** :

$$x_4 = x_7[x_{11}] \tag{38}$$

**F.4** :

$$x_7 = x_9 \# x_{10} \tag{39}$$

$$x_{12} < \mathrm{len}[x_6] \tag{40}$$

$$x_{11} = x_{12} + 1 \tag{41}$$

Here, $x_{10}[x_{12}]$ refers to the $x_{12}{}^{\mathrm{th}}$ element in the list $x_{10}$; len is the length function on a list; $x_9 \# x_{10}$ is a list where the element $x_9$ is concatenated to the front of the list $x_{10}$. **F.2** ⇒ **F.1** by instantiating the quantified variable $x$ in (35) to $x_{12}$ and combining with (36 - 37). **F.1**, **F.4** ⇒ **F.3**:

$$x_4 = x_{10}[x_{12}] = (x_9 \# x_{10})(x_{12} + 1) = x_7[x_{11}].$$

