# OpenReview forum: "IsarStep: a Benchmark for High-level Mathematical Reasoning"
_ICLR.cc/2021/Conference — ICLR 2021 Poster_

### Official Review · AnonReviewer2 · 2020-10-25
**IsarStep: a Benchmark for High-level Mathematical Reasoning**

**Rating:** 6
**Confidence:** 4

**Review:**

The authors propose a new benchmark task to evaluate the high-level reasoning capabilities of machine learning models (specifically sequence-to-sequence models) in the context of proof assistants. The task consists of predicting the intermediate proposition from its surrounding ones, namely its previous and its subsequent propositions. The experimental analysis provides evidence on the difficulty of the task at hand. The authors propose also a solution based on a hierarchical transformer, which is able to better capture the mathematical relations of intra- and inter-propositions compared to existing sequence-to-sequence models, as demonstrated by quantitative as well as qualitative analyses.

The paper is clearly written and has a good balance between technicality and readability.
Provided examples are pedagogical to better understand the introduced concepts. Also, it is positive the fact that the authors are prone to publish their data and code to foster the reproducibility of the experiments.

The major drawback with the paper is in the weak/not well supported motivations of the proposed benchmark task. Also, a discussion of the differences between the proposed model and existing hierarchical transformer architectures is missing. Please, refer below to more detailed comments.

Taking into account that it's not clear to me why the proposed benchmark task is necessary to advance the research in the field of proof assistants, I consider the paper marginally below the acceptance threshold and therefore recommend for an initial rejection. Nevertheless, I'm willing to raise my score if the authors can provide a better explanation on their motivations or provide more convincing arguments supporting the need of their proposed benchmark task. Furthermore, I suggest the authors to discuss some missing related work on hierarchical transformers.

DETAILED COMMENTS

The authors argue that "solving the IsarStep task will be potentially helpful for improving the automation of theorem provers, because proposing a valid intermediate proposition will help to reduce their search space significantly". In general, I agree with the authors that developing benchmarks is an essential driving factor in research and that designing methods able to reduce the search space is essential to improve the automation of theorem provers. I'm not able to see why and how IsarStep can drive this advancement though.
Proofs, both procedural and declarative ones, are inherently sequential and IsarStep breaks this sequentiality by assuming that the proposition subsequent to the missing one is given. For instance, consider the same example used in Section 2 to prove the irrationality of the square root of 2. Why can statement (3) be considered given and in which practical situations does the task of predicting (2) given (1) and (3) occur? Does the IsarStep task occur in practice when proving new conjectures? Wouldn't it be more natural to predict (2) and subsequently (3) by having only (1)?

Furthermore, in which sense is solving IsarStep "a first step towards the long-term goal of sketching complete human-readable proofs automatically"? Can you elaborate more on that?

Hierarchical transformers have been already proposed in natural language for the purposes of document summarization [1-2]. Can you relate with these existing works and particularly discuss what are the architectural novelties of your proposed transformer, as this is one of the contributions listed in the introduction?

MINOR COMMENTS

In the experimental section regarding the visualisation of attention, can you specify what is F2 and what is F3?

[1] Zhang et al. HIBERT: Document Level Pre-training of Hierarchical Bidirectional Transformers for Document Summarization. ACL 2019
[2] Liu and Lapata. Hierarchical Transformers for Multi-Document Summarization. ACL 2019

#########################

UPDATE

Authors have clarified the doubts raised by my questions. I believe that the task proposed in the paper provides new insights on the weaknesses of deep learning models. Therefore, solving the task is important to advance the automation of proof assistants through machine learning. Based on this, I recommend for acceptance.

---

> ### Author Response · Authors · 2020-11-15
> **Author response 1**
>
> Thank you for your thoughtful review.
>
> **Non-sequential proving**: We agree that both procedural/tactic and declarative proofs appear sequential in the surface form of complete proofs, i.e., a sequence of tactic applications in procedural proofs or intermediate propositions in declarative ones. However, we argue that devising those proofs is not really a sequential process. Even in tactic proofs, the users of proof assistants are constantly required to conjecture and prove auxiliary lemmas before proceeding with tactics. When developing declarative proofs, the process is usually more non-sequential. For example, starting with the initial state (e.g., to drive False given 2 is rational) we are likely to sketch/conjecture a few key intermediate propositions first (e.g., a b are coprime, a is even, and a is even) that potentially reduce the gap between ‘2 is rational’ and ‘False’. We then attempt to close the derivation by invoking ATPs. If it fails, we elaborate the derivation with more intermediate steps so that the gap between consecutive ones becomes smaller. We conjecture that bridging large gaps (e.g., 2 is rational -> False) is harder than smaller ones (e.g., 2 b^2 = a^2 -> \exists c. a = 2 c), so we start with building a relatively easy benchmark on small gaps -- IsarStep, which already appears challenging to existing models.
>
> **Example in Section 2**: Back to the example in Section 2, predicting (2) should not only condition on (1) but also a step along the follow-up derivation, otherwise (2) is merely a valid derivation from (1) without a clear goal. Among all the steps along the derivation including ‘b is even’ and the eventual goal ‘False’, we believe that the immediate next step (3) is the easiest condition (3) to solve the synthesis task successfully, hence we use it as a starting point for benchmarking proposition synthesis.
>
> **Towards sketching human-readable proofs**: We say that IsarStep is a step towards sketching human-readable proofs because
> 1. The mined proofs are declarative proofs, a proof style very close to human prose proofs. The proofs are legible, even to people who are not familiar with the system, and they capture high-level structures like those in human proofs.
> 2. A prerequisite for generating a complete proof is the ability to generate a single step of a proof given sufficient conditions. We, therefore, start from such a simpler problem before we investigate whether it is possible for an agent to generate complete human-readable proofs. We provide sufficient condition for the single step of a proof by providing surrounding proofs and letting the agent fill in the gap of proofs. If agents have difficulty in solving our defined task, then it is impossible for them to generate complete proofs.
>
> In particular for Reason 1, the declarative proofs are very different from the previous tactic synthesis frameworks, such as HOList, CoqGym, and TacticToe, where the agent attempts to select a tactic (from a predefined set whose size is usually less than 40) and some arguments (e.g., a pointer to a previously proved lemma) for each proof state. A synthesised tactic proof can be accepted by a proof assistant but is not legible and appears far different from the informal one (which in a sense is a sequence of intermediate propositions such as ‘2 b^2 = a^2’ and ‘a is even’).
>
> **Hierarchical models**: We have added a paragraph relating hierarchical models in the related work section. We also include the discussion here. Hierarchical models have been proposed to solve natural language processing tasks such as document representation (Yang et al., 2016) and document summarisation (Zhang et al.,2019; Liu & Lapata, 2019). Both our hierarchical transformer (HAT) and those models share the similar spirit of introducing local layers to encode local sentences (or propositions) and global layers to capture cross sentence (or proposition) information. However, our HAT is different from their hierarchical models in the way of representing sentences (or propositions): while their models encode sentences into fixed-size vectors, the representation of a proposition in our model is a matrix of dynamic size. The model by Liu & Lapata (2019) has a more sophisticated architecture for capturing sentence representations compared to those by Yang et al. (2016) and Zhang et al. (2019), where they introduce multi-head pooling to encode sentences with different attention weights. Compared to Liu& Lapata (2019)’s model, our model does not introduce additional parameters beyond the standard transformers. Another subtle difference between our model and the existing models is the way of doing positional encoding. Unlike documents where the order of sentences matters, propositions within each category of our IsarStep task do not require an order. Therefore, we do not encode the positional information of different propositions.

---

> > ### Comment · AnonReviewer2 · 2020-11-20
> > **Points clarified by the authors**
> >
> > Thank you for addressing and clarifying my points.
> > I found the argument about the non-sequentiality nature of theorem proving convincing. Originally, this was my most important concern. Also, it's good to see that you have discussed the differences of your model with existing hierarchical transformers.
> >
> > Based on this, I'm happy to increase my score and recommend for acceptance.

---

> > > ### Author Response · Authors · 2020-11-21
> > > **Thank you**
> > >
> > > Many thanks again for your efforts and suggestions. They really have helped us improve the paper.

---

> ### Author Response · Authors · 2020-11-15
> **Author response 2**
>
> **What is F.2 and what is F.3**: In the example of the visualisation of attention, the source is F.2: F.3:$x_{70} \in x_{39}$ F.4:$x_{57} \subseteq x_{39}$, where F.2 is the set of used local propositions (which is empty here), F.3 is the goal we wish to derive (here it is $x_{70} \in x_{39}$), and F.4 is the additional set of propositions we can use to derive F.3 (here it only contains $x_{57} \subseteq x_{39}$). Therefore, the source can be interpreted as ‘In conjunction with $x_{57} \subseteq x_{39}$, what do we need to derive $x_{70} \in x_{39}$?’, and the target answer is $x_{70} \in x_{57}$.

---

### Official Review · AnonReviewer1 · 2020-10-26
**The largest problem is that the dataset is said to be available, but the link is expressed as xxxx.**

**Rating:** 7
**Confidence:** 5

**Review:**

This paper presents a non-synthetic dataset generated from the Isabelle AFP, the largest mechanised proof repository for the task of filling in a missing intermediate proposition given surrounding proofs. Together with the dataset the paper presents a hierarchical transformer model (HAT). Top-10 accuracy, which is the percentage of target proposition appearing in the top 10 generated propositions, is 37.2 for their HAT model. The task this paper addresses is important for theorem proving practitioners, and synthesising propositions appears to be a more difficult challenge compared to other ML tasks in theorem proving (choosing useful lemmas or choosing tactics).

Strengths:

- The database is produced from a non-synthetic dataset called the Archive of Formal Proofs (AFP). AFP is a good source to produce a database that covers diverse topics.
- The newly proposed hierarchical transformer model outperforms the baseline models.
- The evaluation includes a comparison to models reported by other researchers.
- The paper uses a simple motivating example, which helps readers understand the task.
- The evaluation results reported in Table 1 and Table 2 are good for the difficult task of synthesising propositions for diverse problems from the AFP.

# Weakness
- The link to the dataset and model is missing as well as the link to their "test suite to check the correctness of types and the validity of the generated propositions using automatic theorem provers". This is a serious problem to evaluate the paper. The results presented in Tables 1 and 2 are good for the difficult task of proposition synthesis, and I want to test their model for some problems myself. Without the link this is not possible

- The contribution to the proof automation is surprisingly small. They reported 70 cases out of about 3000 are newly proved given the generated intermediate propositions from their HAT model. From the numbers reported in Table 2, I would expect more improvements could be achieved. Without further investigation it is hard to understand this discrepancy.

# Further comments:

- Maybe, it is better to give short names to F1 ~ F5 instead of Fs and numbers?  For example, F1 can be "target" in bold or in teletype font.
- In Section 3.1, the example token pops out of the column for the main text. Maybe removing spaces after the left parenthesis and before the right parenthesis helps?
- In Section 4, the vector "x" is underlined. Is this aligned with the convention in the community? I saw that "x" with an over-line to denote vectors in the past.
- "as the sequence of source propositions with I propositions" -> "as the sequence of I source propositions"?
- In Section 5.2,  it seems reasonable to consider "alternative valid propositions". Can you formally introduce the definition of "correct proposition"? I guess a "correct proposition" is a proposition that matches either the corresponding ground truth or one of the alternative valid propositions. Am I right?
- In the paragraph for Alternative  Valid Propositions in Section 5.2, "5% more correct propositions" -> "5 percentage point more correct propositions"?
- In Section 5.2, the reported contribution to the automation is surprisingly small considering the numbers reported in Table 1 and Table 2. 70 out of 3000 is a 2.3 percentage point increment, even though the previous paragraph says "alternative propositions contribute 5% more correct positions".  Does this mean that some of propositions produced by HAT are valid intermediate lemmas that can be derived from local propositions (F2) and that can be used to prove the goal (F3), but the ATPs are so strong that they can prove F3 from F2 without the valid intermediate lemmas, i.e. IsarStep tends to include small steps that can be skipped by the state-of-the-art ATPs? It is natural if this happens since ATPs are continuously improved and sometimes Isabelle users intentionally write down small steps that are not necessary for the state-of-the-art ATPs.
- In Table 3 in Appendix B, I guess you forgot to multiply numbers by 100 to compute percentage. If so, these numbers in Table 3 appear to be good to me, and it is interesting to know that Transformer slightly outperforms HAT in Table 3 even though HAT outperforms Transformer for producing "correct propositions".

---

> ### Author Response · Authors · 2020-11-14
> **Author response to the availability of dataset & automation**
>
> Thank you for your thoughtful review.
>
> You can find the dataset, model, and test suite in the supplementary materials. Please let us know if there’s any trouble using them on your platform. We will replace the xxx in the paper with an actual Github link once the anonymous period is over.
>
> As for the small increase in automation, there are two reasons for this. When the gap is small enough, Sledgehammer can automatically prove F.3 without being supplied with F.1. This happens in about one-third of the correct synthesises (i.e., those matching either the corresponding ground truth or one of the alternative valid propositions). For the majority of cases, the gaps in F.2 -> F.1 and F.1 -> F.3 are still too big for Sledgehammer to fill in, so the bottleneck is at the ATPs (even with correctly synthesised propositions). We believe further automation can be achieved if tactic-synthesis frameworks like HOList and TacticToe in HOL4 can be jointly incorporated with Sledgehammer to close the gaps between propositions. At the moment, we are re-running our test suite and will update the paper with more discussions on the automation issue.
>
> Since the paper is mainly focusing on providing a benchmark for synthesising propositions, we don’t agree that the small improvement of automation is a weakness of the paper. We actually believe that it is a good finding showing that existing neural seq2seq models are not able to solve the problem yet and therefore it hopefully will encourage more work focusing on the improvement of automation.
>
> We will address your other comments in the revised version of the paper.

---

> > ### Author Response · Authors · 2020-11-23
> > **Further elaboration regarding the automation improvement**
> >
> > We have added the following paragraph to the appendix. We hope this can, to some extent, explain the small improvement in automation.
> >
> > With such settings, Sledgehammer can automatically prove the goal F.3 in 1984/4908 examples. By incorporating the ground truth F.1, the derivations (i.e., to prove both $F.2 \Rightarrow F.1$ and $F.1 \Rightarrow F.3$) can be closed automatically in 2243 examples. Among the 1125 examples that HAT produces an exact match, 466 of them have a 'small' gap: Sledgehammer discharges the goal F.3 directly; 61 of gaps are 'just right': the introduced intermediate step $F.1$ can help Sledgehammer bridge the gap; most of the remaining 598 examples have 'large' gaps (in either $F.2 \Rightarrow F.1$ or $F.1 \Rightarrow F.3$) that are beyond the capability of Sledgehammer. It appears that the insignificant amount of automation improvement can be attributed to the limited number of 'just right' gaps that are within the reach of Sledgehammer.
> >
> > We have also addressed your further comments in the updated version.
> >
> > Thank you again for your insightful suggestions!

---

> > > ### Author Response · Authors · 2020-11-24
> > > **Does our rebuttal address your concerns?**
> > >
> > > We wonder if our rebuttal have addressed your concerns. If not, we are happy to clarify further. Please let us know. Thank you very much.

---

> > > > ### Comment · AnonReviewer1 · 2020-11-24
> > > > **Score**
> > > >
> > > > Based on your answers and inspecting the dataset, I have increased my score.

---

### Official Review · AnonReviewer3 · 2020-10-27
**Official Blind Review #3**

**Rating:** 9
**Confidence:** 4

**Review:**

##########################################################################

Summary:

This paper proposes a benchmark for high-level mathematical reasoning and study the reasoning capabilities of neural sequence-to-sequence models. This is a non-synthetic dataset from the largest repository of proofs written by human experts in a theorem prover, which has a broad coverage of undergraduate and research-level mathematical and computer science theorems. Based on this dataset, the model need to fill in a missing intermediate proposition given surrounding proofs, named as IsarStep. It's a very interesting task. This task provides a starting point for the long-term goal of having machines generate human-readable proofs automatically. The experiments and analysis also reveal that neural models can capture non-trivial mathematical reasoning.


##########################################################################

Reasons for score:


Overall, I strongly vote for accepting. I think this is a very important work and this benchwork would benefit to other fresh ideas and new approaches for mathematical reasoning related research. My only concern is that as a benchmark, do authors need to conduct more experiments and baseline models on their data set to be more convincing?


##########################################################################Pros:

Pros:

+ 1. The paper mined a large corpus of formal proofs and defined a proposition generation task as a benchmark for testing machine learning models’ mathematical reasoning capabilities. Such beckmark is important and beneficial to the development of the artificial intelligence community.


+ 2. The proposed HAT model is novel for better capturing reasoning between source and target propositions. The design of two types of layers is reasonable and interesting. The local layers model the correlation between tokens within a proposition, and the global layers model the correlation between propositions.


+ 3. This paper provides comprehensive experiments, including both qualitative analysis and quantitative results, to show the effectiveness of the proposed model. Experiments and analysis reveal that while the IsarStep task is challenging, neural models can capture non-trivial mathematical reasoning.


+ 4. The paper is well-written and the design decisions are clearly explained. The comparison of benchmark methods is also interesting to read. In general, I think this is a worthy publication.


##########################################################################

Cons:

My only concern is that as a benchmark, do authors need to conduct more experiments and baseline models on their data set to be more convincing? The authors only use two baseline models: RNNSearch and transformer, it seems to be insufficient. At Bert era, do those improved models based on Generative Learning tasks could also be applied to IsarStep as baseline, like MASS(Masked Sequence to Sequence Pre-training for Language Generation)/UNILM(Uniﬁed Language Model Pre-training for Natural Language Understanding and Generation)?

---

> ### Author Response · Authors · 2020-11-15
> **Author response**
>
> Thanks for liking the paper and thanks for your kind feedback.
>
> We have run the convolutional seq2seq model [1] as another baseline model. Please refer to Table 3 in the revised version of the paper for additional results. Now we have results from the three major types of neural networks: the convolutional model, RNN, and the transformer.
>
> Applying pretrained models on the Isarstep dataset is an interesting idea. We have tried pretraining the MASS model on the propositions of our training data following the open source directory (https://github.com/microsoft/MASS/tree/master/MASS-summarization). Our initial experimental results on MASS so far aren't great. We got 4% accuracy on our test set -- there seems to be a bug somewhere. We are still investigating the problem. In the meantime, we found that some concurrent work has explored the idea of pretraining on theorem proving. For example, Polu & Sutskever [2] show that the GPT-3 language model additionally pretrained with mathematical equations mined from the web can generate propositions that enable theorem provers to prove more theorems automatically. Rabe et al. (2020) [3] pretrain masked language models on proofs mined from the HOList dataset and apply the pretrained models to the downstream tasks of type inference and predicting conjectures. We will leave the investigation of the effectiveness of different pretraining methods and different source of pretraining data for IsarStep for future work.
>
> [1] Jonas Gehring, Michael Auli, David Grangier, Denis Yarats, Yann N. Dauphin, Convolutional Sequence to Sequence Learning, ICML 2019.
>
> [2] Stanislas Polu, Ilya Sutskever, Generative Language Modeling for Automated Theorem Proving, arXiv, 2020.
>
> [3] Markus N Rabe, Dennis Lee, Kshitij Bansal, and Christian Szegedy.  Mathematical reasoning viaself-supervised skip-tree training.arXiv preprint arXiv:2006.04757, 2020.

---

### Official Review · AnonReviewer4 · 2020-10-30
**Nice dataset, not quite novel**

**Rating:** 6
**Confidence:** 4

**Review:**

The paper introduces a dataset for proposing intermediate
lemmas/conjectures based on a repository of Isabelle
formalizations. It also does an evaluation of neural
sequence-to-sequence methods on the dataset complemented to some
extent by ATP evaluation. A hierarchical transformer is proposed that
outperforms a transformer baseline.

This is a useful task and such datasets are useful. Apart from the
dataset being based on Isabelle (which is indeed a major proof
assistant with a nice library), there is however not much
novelty. Similar datasets have been extracted from other libraries and
neural experiments have been done over them. To those cited, I would
add [1] based on the large E_conj dataset [2] created in 2017.
Compared to the dataset presented here, [2] also provides an effective
computational metric (based on hundreds of thousands of E prover runs)
for judging the usefulness of the intermediate lemmas. Such datasets
are easy to produce and to scale up to orders of magnitude larger
sizes.

Similar works also include AIMLEAP [3] and the datasets created by
Piotrowski using seq2seq for predicting the next tableau steps [4].

One big issue with the dataset compared to the related work is that it
is not clear how to run the ATP evaluation. The interesting parts of
combining ML and TP typically occur when one can easily run tools on
both sides, do feedback loops, etc.

On the other hand, the ML study seems interesting, it is good that the
Isabelle library is starting to be used this way, and the hierarchical
transformer seems to help. Hence my neutral to mildly positive score,
even though the claims about the uniqueness of the benchmark should be
corrected.


Some more detailed remarks:

p2: give a pointer to declarative vs procedural proofs - the sqrt 2 example has been largely developed by Wiedijk and the declarative proof style comes from Mizar

p4: Free variable normalization
==>
treating of (eigen)variables has been a big topic in the ML-for-TP area. For the most recent feature-based encodings see e.g. the ENIGMA approach. For principled neural treatment, see e.g. the work of Olsak on invariant GNNs [3].

p6: how is the ATP evaluation done?

References:

[1] Zarathustra Goertzel and Josef Urban: Usefulness of Lemmas via Graph Neural Networks.
http://aitp-conference.org/2019/abstract/AITP_2019_paper_32.pdf

[2] https://github.com/JUrban/E_conj

[3] Learning to Advise an Equational Prover
Chad E. Brown, Bartosz Piotrowski, and Josef Urban
http://aitp-conference.org/2020/abstract/paper_32.pdf

[4] Bartosz Piotrowski, Josef Urban:
Guiding Inferences in Connection Tableau by Recurrent Neural Networks. CICM 2020: 309-314

[5] Miroslav Olsák et al:
Property Invariant Embedding for Automated Reasoning. ECAI 2020: 1395-1402

================

UPDATE

Thanks to the authors for their replies and paper updates. My overall evaluation remains on the slightly positive side: I believe that conjecturing is an important task and Isabelle provides a nice corpus for that. Even if declarative proof corpora based on Mizar have been used for similar ML/TP tasks before and the work is not quite novel.

Further notes:

- I would recommend exporting the corpus in the TPTP format. This typically makes the ATP evaluation and building of ML/ATP feedback loops (much) more accessible to ATP researchers, allows including the benchmark in the CASC LTB (large-theory batch) competition, etc. This has been done before [6] for the large corpus of declarative Jaskowski-style Mizar proofs that can be easily used in a similar way as the Isabelle data provided here.

- I do not quite agree with the response claim that E_conj is synthetic and focused on ranking and classification. It is derived from a real-world (Mizar) problem set and the ATP-synthesized lemmas are equipped with a metric of their real-world usefulness. So the various tasks such as regression, ranking, classification and (indeed) synthesis (there is nothing hard about synthesis in the E_conj scenario) have a direct impact in terms of suitable splitting of the real-world ATP problems and their easier solution. It is the same cut introduction task as the authors consider here, just with much more data derived from ATP runs and their characteristics rather than from human proofs. This kind of ATP-based data augmentation is one of the most useful ones in the ML-for-TP domain - quite often more useful than working with human proofs [7] because the ultimate evaluation scenario typically involves ATPs. So it is certainly not the kind of artificial/synthetic task that has an unclear real-world value.

[6] Josef Urban, Geoff Sutcliffe: ATP-based Cross-Verification of Mizar Proofs: Method, Systems, and First Experiments. Math. Comput. Sci. 2(2): 231-251 (2008)

[7] Daniel Kühlwein, Josef Urban: Learning from Multiple Proofs: First Experiments. PAAR@IJCAR 2012: 82-94

---

> ### Author Response · Authors · 2020-11-15
> **Author response**
>
> Thank you for your thoughtful review.
>
> **Related Work**: We appreciate the pointer to the additional related papers, and have cited & discussed them in our related work section. However, we argue that although proposition synthesis (or conjecturing) has been investigated before, most of them are derived from templates in a rule-based fashion. And there are few datasets where the benchmarked models are required to synthesise propositions from scratch from a big vocabulary of 30K mathematical tokens. The most related one is [0], which was cited and compared within our paper. In contrast, we believe both [1] and [3]  focus on classification/ranking tasks. A notable feature of our dataset is that it is non-synthetic: each target proposition in our dataset is from a human input rather than a step in automatic theorem proving. We believe that intermediate steps extracted from proofs produced by saturation-style/connection-based automatic theorem provers such as those from [2] and [4] are unlikely to reflect human-like/human-level conjecturing, which is what we want to model and benchmark in IsarStep.
>
> **ATP evaluation**: We have defined diagnostic commands (i.e.,  ‘check_derivation’ and ‘check_derivation_C’ in EvaluationSuite/Isabelle2019_adapted/src/HOL/Sequence_Evaluation.thy), which take output sequences from neural models as arguments. These diagnostic commands will be inserted to appropriate positions of the corresponding Isabelle theory file, and this modified theory file will be built using Isabelle binaries. Upon building, the diagnostic command will attempt to parse the sequence to a valid Isabelle term F.1’. If succeeded, the command will subsequently call Sledgehammer (with the default 30s timeout) to check the derivations like F.2 -> F.1’’ and F.1’ -> F.3. More details can be found in the source code of our test suite, which is available from the submitted supplementary material.
>
> **Sqrt2 example**: We have added a pointer to Wiedijk’s comparison [5] and the Mizar system, but the sqrt2 example in our paper was independently developed by us and was largely different from the one presented in Wiedijk’s book.
>
> [0] Josef Urban and Jan Jakubuv. First neural conjecturing datasets and experiments. In Christoph Benzmüller and Bruce R. Miller (eds.), Intelligent Computer Mathematics, 2020.
>
> [1] Zarathustra Goertzel and Josef Urban: Usefulness of Lemmas via Graph Neural Networks. http://aitp-conference.org/2019/abstract/AITP_2019_paper_32.pdf
>
> [2] https://github.com/JUrban/E_conj
>
> [3] Learning to Advise an Equational Prover Chad E. Brown, Bartosz Piotrowski, and Josef Urban http://aitp-conference.org/2020/abstract/paper_32.pdf
>
> [4] Bartosz Piotrowski, Josef Urban: Guiding Inferences in Connection Tableau by Recurrent Neural Networks. CICM 2020: 309-314
>
> [5] Freek Wiedijk. The seventeen provers of the world. Springer, 2006.

---

> > ### Author Response · Authors · 2020-11-24
> > **Does our rebuttal address your concerns?**
> >
> > We wonder if our rebuttal have addressed your concerns. If not, we are happy to clarify further. Please let us know. Thank you very much.

---

### Decision · Program_Chairs · 2021-01-07
**Final Decision**

**Decision:**

Accept (Poster)

**Comment:**

This paper introduces a clever new problem that may prove useful in the advancement of Automatic Theorem Proving -- finding intermediate steps in a proof. A non-synthetic benchmark is created based on a large human-created dataset of proofs. Neural models were shown to have non-trivial performance. Reviewers were convinced that this is ultimately a useful benchmark.